# U-shaped association between waist-to-height ratio and microalbuminuria: A cross-sectional analysis conducted within the Chinese demographic

Xia Huang[1‡], Haofei Hu[2‡], Lishu He[3‡], Zhichao Zhang[1], Xue Zhang[1], Hao Yuan[1], Xiaofang He[1], Lirong Tu[1], Feiyuan Liu[4], Xiuqin Li[5], Heping Zhang[1*], Yongcheng He[1*]

**1** Department of Nephrology, Affiliated Hospital of North Sichuan Medical College, Nanchong, Sichuan, China, **2** Department of Nephrology, The First Affiliated Hospital of Shenzhen University, Shenzhen, Guangdong, China, **3** Department of Physiology, University of Arizona College of Medicine - Tucson, Tucson, Arizona, United States of America, **4** Department of Nephrology, Langzhong Hospital of Traditional Chinese Medicine, Langzhong, Nanchong, Sichuan, China, **5** Department of Science and Education, Shenzhen Hengsheng Hospital, Shenzhen, Guangdong, China

‡ These authors are co-first authors.
* 867801115@qq.com (HZ); heyongcheng640815@126.com (YH)

## Abstract

### Background

While previous studies have largely overlooked the potential correlation between the waist-to-height ratio (WHtR) and microalbuminuria in Chinese adults, this study aims to rigorously investigate this relationship. Specifically, we examined the potential non-linear association between WHtR and the presence of microalbuminuria.

### Methods

We conducted a cross-sectional analysis of 33,685 participants from eight regions across China. Microalbuminuria was defined as a urinary albumin-to-creatinine ratio (ACR) > 30 mg/g. The relationship between WHtR and microalbuminuria was assessed using univariate and multivariate logistic regression models. We further explored potential nonlinear associations using Generalized Additive Models (GAM) and evaluated threshold effects to identify critical inflection points. Subgroup analyses were performed to validate the robustness of our findings.

### Results

The cohort (N = 33,685) had a mean age of 57.6 ± 9.27 years and was predominantly females (22,516; 66.8%). The mean WHtR was 0.537 ± 0.06. The median ACR value observed was 9.93 mg/g with an interquartile range (IQR) of 8.23–11.68 mg/g. Notably, the prevalence rate of microalbuminuria detected in the study was 14.4%. In multivariate-adjusted models, each one-unit increase in WHtR was associated with a

⛓ OPEN ACCESS

**Data availability statement:** The minimal dataset underlying the findings of this study is publicly available in the figshare repository at: https://doi.org/10.6084/m9.figshare.31404420.

**Funding:** No. JCYJ20210324133412033 the Shenzhen Science and Technology Innovation Committee. the Shenzhen Municipal Health Commission (Grant No. SZXJ2017031).

**Competing interests:** The authors have declared that no competing interests exist.

**Abbreviations:** BMI, Body Mass Index; WC, Waist Circumference; HC, Hip Circumference; WHR, Waist-to-Hip Ratio; WHTR, Waist-to-Height Ratio; SBP, Systolic Blood Pressure; DBP, Diastolic Blood Pressure; FPG, Fasting Plasma Glucose (equivalent to Fasting Blood Glucose); PPG, Postprandial Plasma Glucose (equivalent to Postprandial Blood Glucose); HbA1c, Glycated Hemoglobin A1c; IR, Insulin Resistance (as assessed by the Homeostatic Model Assessment of Insulin Resistance or other validated indices); TC, Total Cholesterol; TG, Triglycerides; HDL-c, High-Density Lipoprotein Cholesterol; LDL-c, Low-Density Lipoprotein Cholesterol; ALT, Alanine Aminotransferase; AST, Aspartate Aminotransferase; GGT, Gamma-Glutamyl Transferase (replacing "Transpeptidase" for standardization); Scr, Serum Creatinine; BUN, Blood Urea Nitrogen; eGFR, Estimated Glomerular Filtration Rate (calculated using the Chronic Kidney Disease Epidemiology Collaboration equation or similar); ACR, Albumin-to-Creatinine Ratio; CKD, Chronic Kidney Disease (staged according to KDIGO guidelines); OR, Odds Ratio; CI, Confidence Interval; Ref, Reference Category (the baseline comparator in regression modeling); SD, Standard Deviation; GAM, Generalized Additive Model (a non-parametric regression technique for exploring non-linear associations).

48.3% higher likelihood of microalbuminuria development (odds ratio [OR] = 1.483; 95% confidence interval [CI]: 1.410–1.559; $p < 0.0001$). GAM analyses revealed a significant non-linear association between WHtR and microalbuminuria ($p < 0.0001$). Segmented logistic regression identified a WHtR threshold value of 0.497. Above this threshold, the risk of microalbuminuria increased markedly (OR = 11.9, 95% CI 5.14–27.56, $p < 0.0001$), whereas below this threshold, WHtR was inversely associated with microalbuminuria (OR = 0.107, 95% CI: 0.015–0.748, $p = 0.0243$). Subgroup analyses further indicated a stronger association among individuals without hypertension and those with a body mass index (BMI) ≥ 24 kg/m², while the association was attenuated in hypertensive participants and those with a BMI < 24 kg/m².

## Conclusions

The risk of microalbuminuria exhibited a U-shaped pattern across WHtR values, with elevated risk at both low and high ends of the spectrum. Moderate central adiposity appeared to confer a protective effect against renal dysfunction. These findings highlighted the dual risks posed by underweight and excessive abdominal obesity and underscored the importance of targeted strategies to prevent obesity-related kidney damage. Future research is warranted to elucidate the mechanisms by which excess visceral fat contributes to renal impairment and to guide interventions aimed at preserving kidney health in diverse populations.

## Introduction

Chronic kidney disease (CKD) is characterized by either a reduction in estimated glomerular filtration rate (eGFR) value to < 60 mL/min/1.73 m² or an ACR of > 30 mg/g, according to established diagnostic guidelines [1]. It affects over 10% of global population [2], CKD represents a major public health challenge. Microalbuminuria, an early and sensitive marker of CKD progression, is also a validated predictor of declining renal function and heightened cardiovascular morbidity risk [3–5], reflecting both glomerular and tubular damage [3].

Amidst the global obesity epidemic, which now affects nearly 2 billion adults worldwide [6], excess adiposity emerged as a modifiable risk factor for CKD. Among various measures of body fat, the waist-to-height ratio (WHtR) exhibits superior predictive value for metabolic comorbidities compared to the conventional body mass index (BMI), as shown in multiple empirical studies [7]. Notably, Asian populations tend to accumulate visceral fat at significantly lower WHtR threshold than their Caucasian counterparts [8].

However, critical knowledge gaps remain regarding the association between WHtR and microalbuminuria. First, most existing studies have assumed a linear relationships [9], despite biological evidence suggesting that adiposity may exert threshold effects on renal hemodynamics and supporting the plausibility of a nonlinear relationship [10]. Second, the commonly cited universal WHtR cutoff of 0.5 [11] has not been validated in Asian

populations, whose body composition and renal risk profiles may require different thresholds [12]. Third, previous studies in Asian cohorts have often been limited by small sample sizes (N<5000), restricting their ability to detect nonlinear patterns [13].

To address these gaps, we conducted a large-scale cross-sectional study of 33,685 Chinese adults. Using restricted cubic spline modeling and comprehensive adjustment for metabolic confounders (e.g., insulin resistance, inflammatory markers), we aimed to characterize the potential nonlinear association between WHtR and microalbuminuria and help establish population-specific WHtR thresholds for early identification of renal risks in clinical practice. While the REACTION study database has been used in previous epidemiological investigations, to the best of our knowledge, this is the first study specifically focusing on the nonlinear, U-shaped relationship between waist-to-height ratio and microalbuminuria. Previous research has primarily examined linear associations or other obesity indices, making our study a unique contribution to the field.

## Methods

### Study design

This study analyzed data from a publicly available dataset originally curated by Ye et al. (https://doi.org/10.1371/journal.pone.0214776) [14]. Waist-to-height ratio (WHtR) was examined as the primary predictor variable, and microalbuminuria was assessed as the primary outcome.

### Data source

Data were obtained from the PLOS ONE publication by Ye et al. [14], which investigated the relationship between sleep patterns and renal function in a healthy Chinese population. The dataset used in this secondary analysis is deposited in the figshare repository and is freely available for non-commercial research under the journal's open access policy and Creative Commons licensing (https://doi.org/10.6084/m9.figshare.31404420). Use of the data is permitted with appropriate citation of the original source.

### Ethics statement

The study protocol was executed in strict adherence to the ethical principles and guidelines delineated in the Declaration of Helsinki. The original study received ethical approval from the Institutional Review Board (IRB) of Rui-Jin Hospital, affiliated with Shanghai Jiao Tong University School of Medicine. Prior to study enrollment, all participants provided written informed consent for voluntary participation in accordance with institutional and national ethical research guidelines.

### Study population

Participants were enrolled as part of the REACTION study—a large, multicenter cohort designed to investigate chronic disease risk factors, including diabetes and cancer, among Chinese adults. A total of 33,850 individuals were recruited from eight geographically diverse urban centers: Shanghai, Guangzhou, Zhengzhou, Dalian, Lanzhou, Luzhou, Wuhan, and Guangxi, ensuring broad regional representation.

Exclusion criteria were applied to remove individuals with (1) known primary renal pathology, (2) regular use of angiotensin-converting enzyme inhibitors (ACEIs) or angiotensin receptor blockers (ARBs), and (3) implausible self-reported sleep durations (< 4 hours or > 12 hours per day). In alignment with the original study protocol, an additional 165 participants (0.5%) were excluded for having extreme WHtR values (> 3 standard deviations from the mean) [15]. The final analytic cohort consisted of 33,685 individuals. The overall study design and participant selection flow were detailed in **Fig 1**.

### Variables

**Independent variable.** Waist-to-height ratio (WHtR) was the primary independent variable and was treated as a continuous numerical variable. WHtR was calculated by dividing each participant's waist circumference (in centimeters) by their standing height (in centimeters).

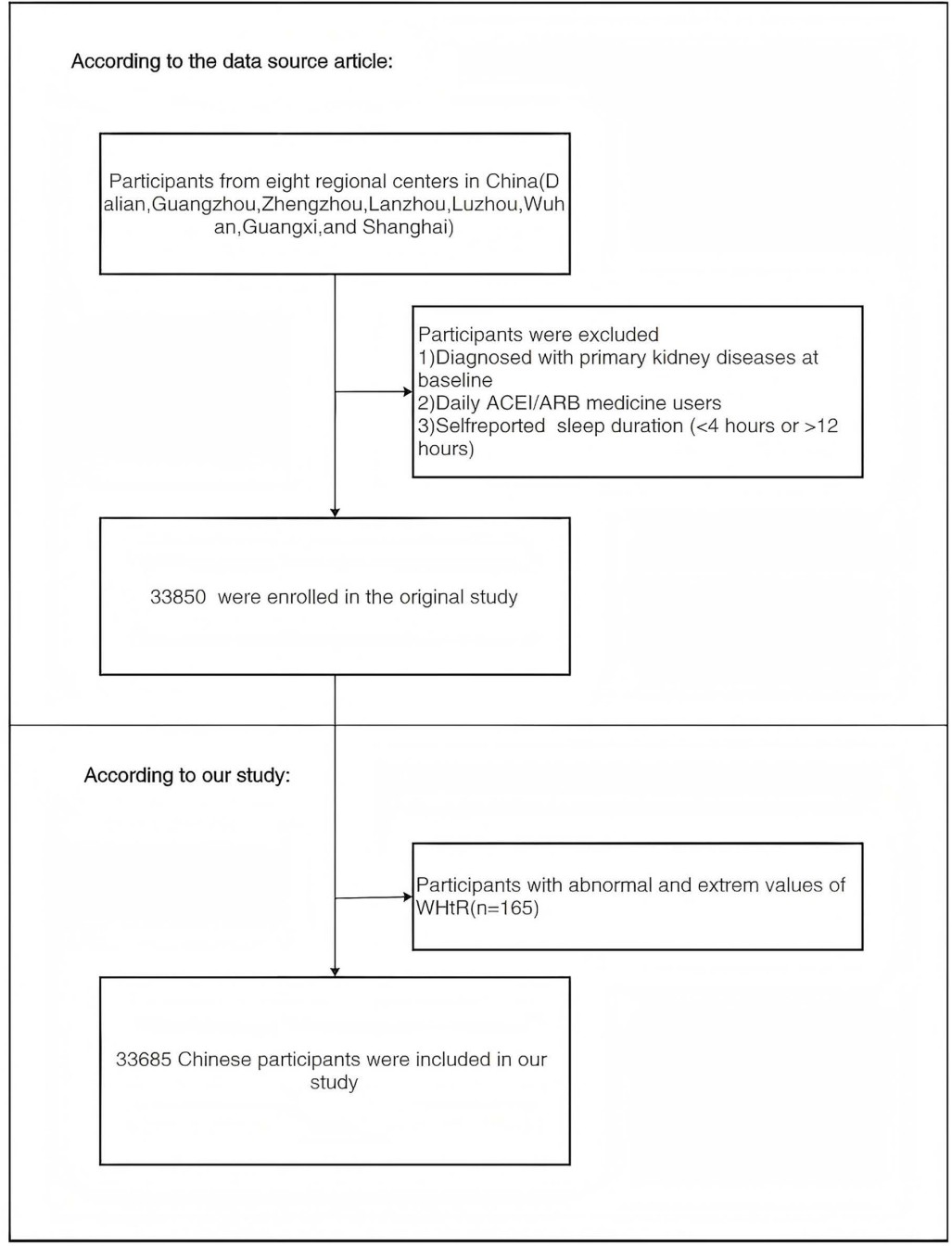

**Fig 1. Study participant selection flowchart.** The diagram illustrates the inclusion/exclusion process. From 33,850 eligible individuals, 165 were excluded, yielding a final analytical sample of 33,685 participants.

**Dependent variable.** The primary outcome of interest was the presence of microalbuminuria, defined as a urinary albumin-to-creatinine ratio (ACR) ≥ 30 mg/g. The variable was operationalized as binary: participants with ACR ≥ 30 mg/g were classified as having microalbuminuria (coded as 1), while those with ACR < 30 mg/g were classified as not having microalbuminuria (coded as 0).

**Covariates.** Study variables were selected based on three key considerations: (1) availability in the original dataset, (2) clinical relevance as evaluated by nephrology experts, and (3) previously established associations in the literature [18]. Both categorical and continuous variables were included:

1. Categorical Variables:

   a. Demographics: sex (male/female)

   b. Medical history: diabetes, hypertension, cancer, obesity, CKD

   c. Lifestyle factors: smoking status, alcohol consumption

2. Continuous Variables:

   a. Anthropometrics: age (years), standing height (cm), body mass (kg), waist circumference (WC, cm), hip circumference (HC, cm)

   b. Blood Pressure: diastolic (DBP, mmHg) and systolic (SBP, mmHg)

   c. Lipid Profiles: triglycerides (TG, mg/dL), high-density lipoprotein cholesterol (HDL-c, mg/dL), low-density lipoprotein cholesterol (LDL-c, mg/dL), total cholesterol (TC, mg/dL)

   d. Renal Function: blood urea nitrogen (BUN, mg/dL), serum creatinine (Scr, mg/dL)

   e. Liver Enzymes: alanine aminotransferase (ALT, U/L), gamma-glutamyl transferase (GGT, U/L), aspartate aminotransferase (AST, U/L)

   f. Glycemic Markers: fasting plasma glucose (FPG, mg/dL), 2-hour postprandial plasma glucose (PPG, mg/dL), glycated hemoglobin (HbA1c, %)

**Data collection.** Participant data were obtained through a standardized questionnaire and physical examination protocol. The questionnaire collected information on demographics (sex, age), medical history (e.g., diabetes, hypertension, cancer, obesity, CKD), lifestyle behaviors (smoking, alcoholic use, physical activity), and daytime sleepiness patterns. Anthropometric measurements were performed by trained personnel following standardized procedures. Briefly, participants removed shoes, hats, and coats before measurement. Height was measured to the nearest centimeter using a calibrated digital ultrasound device. Body weight was measured to the nearest 0.1 kg using an electronic scale. Waist circumference was measured at the midpoint between the lower border of the 12th rib and the iliac crest, ensuring that the measuring tape was level and parallel to the floor. Hip circumference was taken at the widest point of the buttocks. WHtR was calculated as waist circumference (cm) divided by height (cm). Body mass index (BMI) was calculated as weight (kg) divided by height (m$^2$): BMI = weight (kg) / [height (m)]$^2$.

**UACR measurements and data processing.** First-morning void urine specimens were obtained after an 8-hour fasting period for quantification of urinary albumin-to-creatinine ratio (ACR). ACR levels were used to assess microalbuminuria status, with classification based on established clinical diagnostic criteria. Microalbuminuria was defined as an ACR ≥ 30 mg/g. Individuals exceeding the 75th percentile of ACR values were categorized as having elevated urinary albumin excretion.

**eGFR calculation.** Estimated glomerular filtration rate (eGFR) was calculated utilizing the Modification of Diet in Renal Disease (MDRD) equation: eGFR (mL/min/1.73 m$^2$) = 186 × [serum creatinine concentration (Scr, in mg/dL)]$^{-1.154}$ × [patient's age (in years)]$^{-0.203}$ × (0.742, if female). CKD was defined as eGFR < 60 mL/min/1.73 m$^2$, while renal hyperfiltration was defined as eGFR > 135 mL/min/1.73 m$^2$.

**Blood pressure assessment protocol.** Following a standardized 5-minute rest in the seated position, blood pressure (BP) measurements were measured three times at 60-second intervals using a validated digital sphygmomanometer. The arithmetic mean of the three readings was used in the final analysis. Hypertension was defined as any of the following: (1) average systolic BP (SBP) ≥ 130 mmHg, (2) average diastolic BP (DBP) ≥ 80 mmHg, or (3) physician-diagnosed hypertension confirmed via self-reported medical history.

**Blood biochemical index measurement.** After an 8-hour fasting period, venous blood specimens were obtained in the morning (between 07:00 and 09:00) using standardized venipuncture techniques. Non-diabetic participants completed a 75-gram oral glucose tolerance test (OGTT), while those with pre-existing diabetes consumed a 100-gram steamed bun within 10 minutes under supervision. Blood was collected at baseline (0 minutes) and 120 minutes post-ingestion.

Comprehensive biochemical profiling included: (1) Lipid metabolism: TG, TC, LDL-c, HDL-c; (2) Renal function: Scr and BUN; (3) Liver function: ALT, AST, GGT; and (4) Glycemic control: FPG, PPG, HbA1c.

**Statistical analysis.** Participants were stratified into WHtR quartiles for comparative analyses: Q1 (0.35–0.50), Q2 (0.50–0.54), Q3 (0.54–0.58), and Q4 (0.58–0.73). Categorical variables were summarized as frequencies and percentages. For continuous variables, descriptive statistics were applied based on data distribution characteristics: normally distributed data were expressed as mean ± standard deviation, while skewed data were summarized as median [interquartile range]. Normality was assessed using the Shapiro-Wilk test.

Between-group comparisons were performed using the χ² test for categorical variables, one-way analysis of variance (ANOVA) for normally distributed continuous variables, and the Kruskal-Wallis test for non-normally distributed variables.

The presence of multicollinearity was assessed via the variance inflation factor (VIF) analysis, with VIF > 5 denoting substantial collinearity, necessitating model refinement and variable exclusion (S1 Table) [16]. VIF was calculated using the formula: $VIF = 1/(1 - R^2)$, where $R^2$ denotes the coefficient of determination derived from the regression of each independent variable against all others.

To assess the association between WHtR and microalbuminuria, three logistic regression models were constructed:

Model I (unadjusted): no covariate adjusted.

Model II (partially adjusted): adjusted for sex, SBP, DBP, tumor status, smoking, alcohol consumption, body weight, and HC).

Model III (fully adjusted): (included all covariates from Model 2, with additional adjustments for ALT, AST, FPG, eFGR, HDL-c, LDL-c, and TG.

Odds ratios (ORs) along with their corresponding 95% confidence intervals (CIs) were reported for each model.

To explore potential nonlinear relationship between WHtR and microalbuminuria, we employed generalized additive models (GAMs) with penalized spline smoothing techniques. Where a nonlinear association was identified, an inflection point was determined through a recursive algorithm. Separate logistic regression models were then fitted for data below and above the identified inflection point, with model selection guided by likelihood ratio testing.

Subgroup analyses were conducted using stratified logistic regression by:

Age: < 60 years of age vs. > 60

Sex: male vs. female

Hypertension status [17]: non-hypertensive (SBP < 140 mmHg and DBP < 90 mmHg) vs. hypertensive (SBP ≥ 140 mmHg or DBP ≥ 90 mmHg)

Diabetes status [18]: non-diabetic (FPG < 7.0 mmol/L and PPG < 11.1 mmol/L, or no diabetes diagnosis) vs. diabetic (FPG ≥ 7.0 mmol/L, PPG ≥ 11.1 mmol/L, or confirmed diabetes diagnosis)

BMI categories [19]: $< 18.5\,kg/m^2$, $18.5 \le BMI < 24\,kg/m^2$, $24 \le BMI < 28\,kg/m^2$, and $BMI \ge 28\,kg/m^2$

Tumor history: tumor vs. non-tumor

All models were adjusted for relevant covariates, including sex, SBP, DBP, alcohol and smoking consumption, tumor history, body weight, ALT, AST, HC, HDL-C, LDL-c, TG, FPG, and eGFR.

Analyses adhered to the STROBE (Strengthening the Reporting of Observational Studies in Epidemiology) guidelines. Statistical computations were performed utilizing EmpowerStats (version 4.1; X&Y Solutions, Inc., Boston, MA; accessible via www.empowerstats.com) and R (The R Foundation; available at www.r-project.org). A two-tailed $p$-value $< 0.05$ was considered statistically significant.

Missing data were identified for lifestyle variables, including smoking (n = 284) and alcohol consumption (n = 268). To optimize statistical power and minimize potential selection bias, participants with missing data were retained and classified as a separate category ('Not recorded') rather than excluded from the analyses. No imputation was performed. The only exclusions related to data quality were the removal of 165 participants (0.5%) with extreme WHtR values (>3 SD from the mean), as detailed in the Methods section. Therefore, no imputation or additional handling of missing data was necessary.

To assess the adequacy of the sample size, we conducted a post-hoc power analysis. Based on the observed prevalence of microalbuminuria (14.4%) in our study population, a post-hoc power analysis indicated that our sample of 33,685 participants provided >99% power to detect an odds ratio of 1.2 at a two-sided alpha level of 0.05. This high statistical power supports the robustness of our findings and indicates that our study was sufficiently powered to identify even modest associations.

## Results

### Demographic and clinical profiling of the study cohort

Table 1 summarizes key sociodemographic characteristics, laboratory biomarkers, and clinical variables collected using standardized protocols. The final study cohort included 33,685 participants, consisting of 11,169 males (33%) and 22,516 females (67%) with a mean age of 57.6 ± 9.25 years. WHtR followed a Gaussian distribution pattern (Fig 2), with a mean of 0.537 ± 0.06. Based on WHtR, participants were stratified into quartiles: Q1 (0.35–0.50), Q2 (0.50–0.54), Q3 (0.54–0.58), and Q4 (0.58–0.73). The median ACR was 9.93 mg/g (interquartile range [IQR]: 8.23–11.68 mg/g), and the mean eGFR was 93.98 ± 19.07 ml/min.

Comparisons between Q4 and Q1 groups revealed that participants in Q4 quartile displayed significantly higher concentrations of HbA1c, BMI, TG, AST, ALT, GGT, LDL-c, CHOL, FPG, PPG, SBP, DBP, WC, HC, age, and WHtR. A higher proportion of individuals in Q4 were female, smokers, drinkers, or had chronic kidney disease, diabetes, or hypertension. In contrast, Q1 participants exhibited elevated HDL-C and eGFR values. No statistically significant differences were found in cancer history across WHtR quartiles. The prevalence of severe obesity increased progressively from Q1 to Q4.

Participants were further grouped based on microalbuminuria status, defined by urinary albumin excretion exceeding the diagnostic threshold. WHtR distributions for both groups are shown in Fig 3. Individuals with microalbuminuria had higher WHtR values than those without.

As illustrated in Fig 4, the prevalence of microalbuminuria increased across WHtR quartiles ($p$<0.001) except for the Q1 group.

Fig 5 demonstrates age- and sex- stratified trends: females showed higher microalbuminuria incidence in all age groups except those < 40 years, and the prevalence increased with age in both sexes (except for males < 40 years).

### Prevalence of microalbuminuria

As shown in Table 2, microalbuminuria was detected in 4,852 participants, resulting in an overall prevalence of 14.40% (95% CI 14.00% to 14.79%) among all individuals. Prevalence by WHtR quartile was: Q1 – 12.04% (11.39%–12.75%), Q2

**Table 1. Baseline characteristics of participants stratified by WHtR quartiles.**

| WHtR quartile | Q1 (0.35–0.50) N = 8420 | Q2 (0.50–0.54) N = 8414 | Q3 (0.54–0.58) N = 8418 | Q4 (0.58–0.73) N = 8433 | p value |
|---|---|---|---|---|---|
| Age (years) | 54.6 ± 8.83 | 56.49 ± 8.57 | 58.18 ± 8.98 | 61.13 ± 9.73 | < 0.001 |
| Height (cm) | 161.62 ± 7.54 | 161.33 ± 7.78 | 160.68 ± 7.86 | 158.01 ± 8.03 | < 0.001 |
| Weight (kg) | 56.12 ± 8.57 | 61.95 ± 8.94 | 65.74 ± 10.01 | 69.93 ± 11.28 | < 0.001 |
| WC (cm) | 74.77 ± 5.46 | 83.24 ± 4.37 | 89.08 ± 4.74 | 97.18 ± 6.31 | < 0.001 |
| HC (cm) | 90.55 ± 5.74 | 95.24 ± 5.56 | 98.47 ± 6.05 | 103.52 ± 7.12 | < 0.001 |
| Scr (umol/L) | 67.96 ± 18.66 | 68.73 ± 15.71 | 69.44 ± 17.20 | 68.03 ± 16.08 | < 0.001 |
| HDL-c (mmol/L) | 1.41 ± 0.36 | 1.31 ± 0.34 | 1.27 ± 0.33 | 1.27 ± 0.31 | < 0.001 |
| LDL-c (mmol/L) | 2.83 ± 0.87 | 2.97 ± 0.90 | 3.02 ± 0.90 | 3.10 ± 0.89 | < 0.001 |
| Chol (mmol/L) | 4.91 ± 1.11 | 5.05 ± 1.15 | 5.11 ± 1.12 | 5.22 ± 1.12 | < 0.001 |
| TG (mmol/L) | 1.09 (0.82-1.52) | 1.35 (0.97-1.94) | 1.49 (1.06-2.14) | 1.59 (1.14-2.24) | < 0.001 |
| ALT (U/L) | 13.00 (10.00-18.00) | 14.00 (11.00-20.00) | 16.00 (12.00-22.00) | 16.00 (12.00-24.00) | < 0.001 |
| AST (U/L) | 20.00 (16.00-24.00) | 20.00 (17.00-24.00) | 20.00 (17.00-25.00) | 21.00 (18.00-26.00) | < 0.001 |
| GGT (U/L) | 17.00 (13.00-25.00) | 20.00 (15.00-30.00) | 22.00 (16.00-34.00) | 24.00 (17.00-38.00) | < 0.001 |
| FPG (mmol/L) | 5.62 ± 1.43 | 5.90 ± 1.66 | 6.09 ± 1.76 | 6.33 ± 1.90 | < 0.001 |
| BPG (mmol/L) | 7.43 ± 3.33 | 8.27 ± 3.83 | 8.79 ± 4.00 | 9.63 ± 4.29 | < 0.001 |
| HbA1c (mmol/L) | 5.87 ± 0.90 | 6.03 ± 1.00 | 6.14 ± 1.07 | 6.33 ± 1.90 | < 0.001 |
| BMI (kg/m$^2$) | 21.42 ± 2.50 | 23.72 ± 2.23 | 25.35 ± 2.49 | 27.89 ± 3.13 | < 0.001 |
| eGFR (ml/min*1.73m$^2$) | 95.78 ± 19.32 | 94.80 ± 18.70 | 93.33 ± 18.63 | 91.99 ± 19.38 | < 0.001 |
| ACR (mg/g) | 9.25 (5.48-17.79) | 9.48 (5.54-18.48) | 10.00 (5.76-20.28) | 11.33 (6.21-23.58) | < 0.001 |
| SBP (mmHg) | 122.67 ± 17.77 | 129.78 ± 19.17 | 134.09 ± 19.56 | 140.61 ± 20.61 | < 0.001 |
| DBP (mmHg) | 74.21 ± 10.17 | 77.26 ± 10.59 | 78.79 ± 10.76 | 80.06 ± 10.98 | < 0.001 |
| WHtR | 0.46 ± 0.03 | 0.52 ± 0.01 | 0.55 ± 0.01 | 0.62 ± 0.03 | < 0.001 |
| Gender | | | | | < 0.001 |
| Male | 2761 (32.79%) | 3095 (36.78%) | 3094 (36.75%) | 2219 (26.31) | |
| Female | 5659 (67.21%) | 5319 (63.22%) | 5324 (63.25%) | 6214 (73.69%) | |
| Tumor | | | | | 0.746 |
| Yes | 247 (2.93%) | 254 (3.02%) | 232 (2.76%) | 252 (2.99%) | |
| No | 8173 (97.07%) | 8160 (96.98%) | 8186 (97.24%) | 8181 (97.01%) | |
| Diabetes Mellitus | | | | | < 0.001 |
| Absent | 7294 (86.63%) | 6748 (80.20%) | 6313 (74.99%) | 5686 (67.43%) | |
| Present | 1126 (13.37%) | 1666 (19.80%) | 2105 (25.01%) | 2747 (32.57%) | |
| Obesity | | | | | < 0.001 |
| Non-obese | 8367 (99.37%) | 8251 (98.06%) | 7536(89.52) | 4545 (53.90%) | |
| Obese | 53 (0.63%) | 163 (1.94%) | 882 (10.48%) | 3888 (46.1%) | |
| CKD | | | | | < 0.001 |
| No | 8279 (98.33%) | 8250 (98.05%) | 8211 (97.54%) | 8123 (96.32%) | |
| Yes | 141 (1.67%) | 164 (1.95%) | 207 (2.46%) | 310 (3.68%) | |
| Hypertension | | | | | < 0.001 |
| Yes | 5103 (60.61%) | 3659 (43.49%) | 2810 (33.38%) | 1840 (21.82%) | |
| No | 3317 (39.39%) | 4755 (56.51%) | 5608 (66.62%) | 6593 (78.18%) | |
| Smoking | | | | | < 0.001 |
| No | 6970 (82.78%) | 6903 (82.04%) | 7054 (83.80%) | 7399 (87.74%) | |
| Yes | 1370 (17.28%) | 1432 (17.96%) | 1311(16.20%) | 964 (12.26%) | |
| Drinking | | | | | < 0.001 |
| No | 6005 (71.32%) | 5969 (70.94%) | 6105 (72.52%) | 6613 (78.42%) | |

*(Continued)*

Table 1. (Continued)

| WHtR quartile | Q1 (0.35–0.50) N=8420 | Q2 (0.50–0.54) N=8414 | Q3 (0.54–0.58) N=8418 | Q4 (0.58–0.73) N=8433 | p value |
|---|---|---|---|---|---|
| Yes | 2342 (28.68%) | 2367 (29.06%) | 2259 (27.48%) | 1760 (21.58%) | |

Categorical variables are expressed as counts (N) and corresponding percentages (%). Continuous variables that are normally distributed are expressed as mean±standard deviation (SD). Continuous variables that deviate from normal distribution are expressed as median (interquartile range).

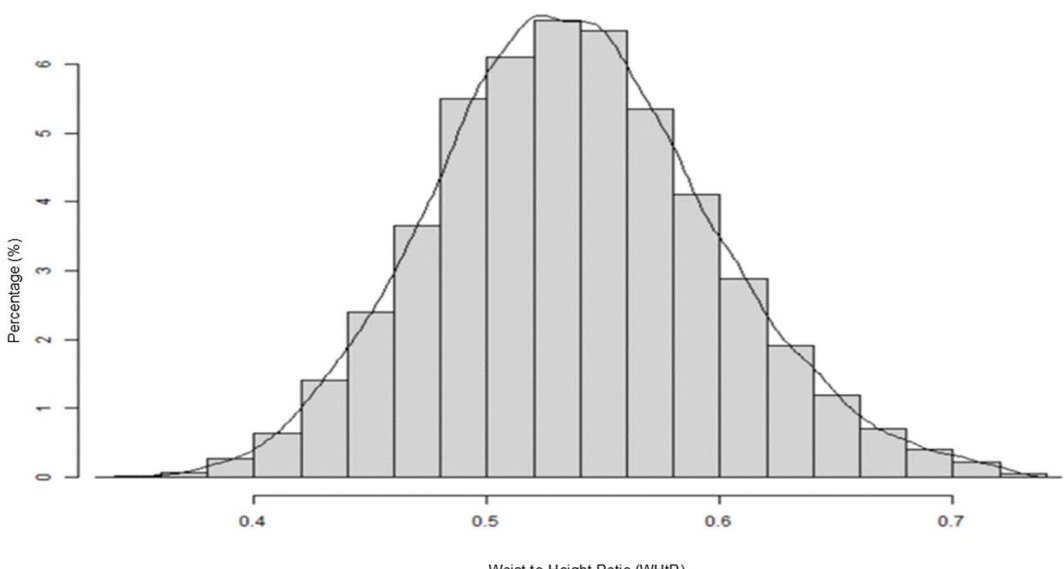

**Fig 2. Distribution of waist-to-height ratio (WHtR).** The histogram demonstrates a normally distributed WHtR across the study population, with values ranging from 0.35 to 0.73.

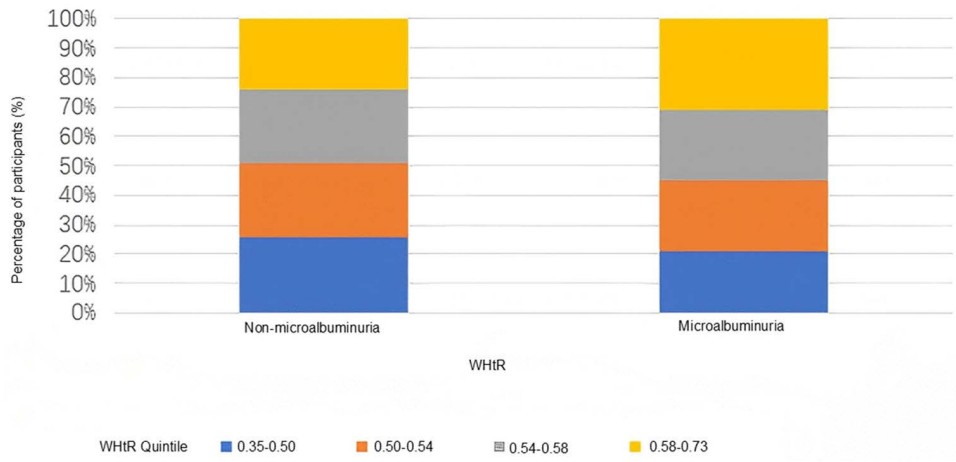

**Fig 3. WHtR distribution by microalbuminuria status.** Participants with microalbuminuria demonstrated significantly higher WHtR values compared to those without microalbuminuria ($p < 0.001$).

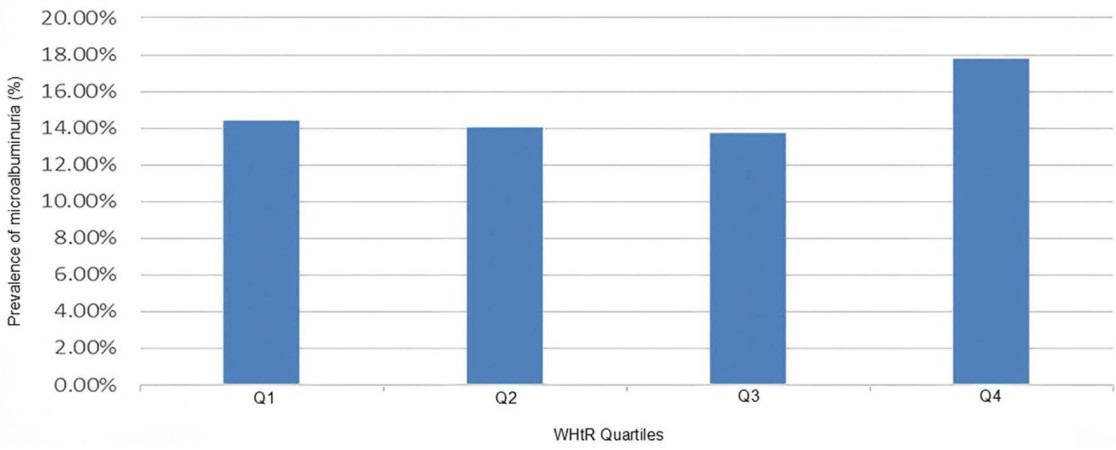

**Fig 4. Microalbuminuria prevalence across WHtR quartiles.** The bar chart illustrates a dose-response association between WHtR quartiles and microalbuminuria prevalence (*p* for trend < 0.001).

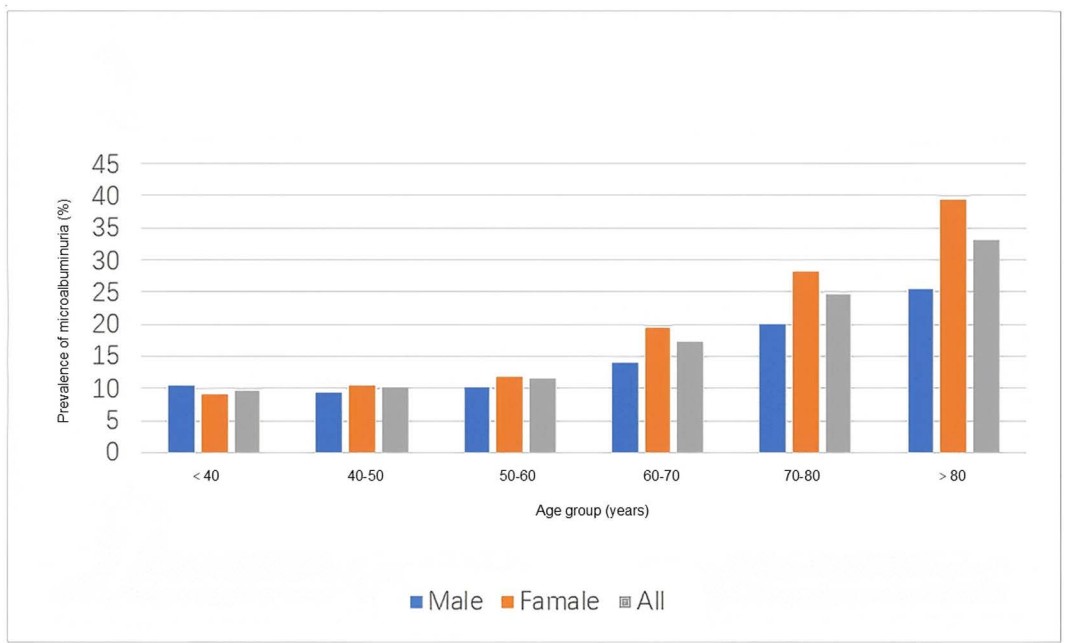

**Fig 5. Age- and sex-specific microalbuminuria prevalence.** Prevalence was compared between males and females across six age groups. Females showed consistently higher prevalence than males in all age groups ≥ 40 years (*p* < 0.05 for each interval).

– 14.00% (13.75%–14.71%), Q3 – 13.75% (17.01%–18.70%) and Q4 – 17.81% (14.00%–14.79%), respectively, indicating a significant upward trend (*p* for trend <0.001).

## Univariate analysis

Univariate analysis (**Table 3**) revealed significant associations between the following and microalbuminuria: AST, ALT, Scr, BMI, FPG, GGT, HbA1c, PPG, TG, DBP, SBP, HC, WC, female sex, age, diabetes, obesity, hypertension, and renal

**Table 2. Distribution of microalbuminuria prevalence by WHtR quartiles along with 95% confidence intervals (CI).**

| WHtR Quartile | Study Cohort (Sample Size: N) | Microalbuminuria Cases (N) | Prevalence Estimate (95% CI) (%) |
|---|---|---|---|
| Total | 33685 | 4852 | 14.40 (14.00-14.79) |
| Q1 (0.35–0.50) | 8420 | 1014 | 12.04 (11.39-12.75) |
| Q2 (0.50–0.54) | 8414 | 1178 | 14.00 (13.27-14.71) |
| Q3 (0.54–0.58) | 8418 | 1158 | 13.75 (12.93-14.54) |
| Q4 (0.58–0.73) | 8433 | 1502 | 17.81 (17.01-18.70) |
| *p* for trend | | < 0.001 | |

insufficiency. Conversely, height, LDL-c, HDL-c, CHOL, and eGFR were negatively associated. Tumor history and body weight were not significantly associated with microalbuminuria. Interestingly, "sometimes" and "regular" smokers or alcohol drinkers showed lower risk compared to "never" users, likely reflecting self-report bias in questionnaire data.

## Multivariate logistic regression

Multivariate-adjusted logistic regression results are summarized in **Table 4**.

Model I (Unadjusted): Each 1-unit increase in WHtR was significantly associated with a 48.3% higher odds of microalbuminuria (OR = 1.483, 95% CI: 1.410–1.559).

Model II (Partially Adjusted): Adjusting for sex, SBP, DBP, tumor status, smoking, alcohol consumption, weight, and HC, the association remained significant (OR = 1.269, 95% CI: 1.183–1.362).

Model III (Fully Adjusted): After adjusting for all potential confounding covariates, each 1-unit increase in WHtR was linked to a 17.3% increase in microalbuminuria prevalence (OR = 1.173, 95% CI: 1.090–1.261).

When WHtR quartiles were entered into the fully adjusted model with Q1 as reference:

Q2: OR = 0.966 (95%CI: 0.874–1.067). 2023.

Q3: OR = 1.028 (95% CI: 0.926-1.141). 2023.

Q4: OR = 1.242 (95% CI: 1.106-1.395).

These results indicated a significant increase in risk only in the highest quartile Q4.

## Nonlinear association between WHtR and microalbuminuria

Using GAM with penalized spline smoothing, a nonlinear, U-shaped relationship between WHtR and microalbuminuria risk was observed (**Fig 6**).

A threshold effect was identified at WHtR = 0.497 via recursive partitioning. Dual-segment logistic regression showed right of threshold (≥ 0.497) with OR = 11.902 (95% CI: 5.141–27.556) and left of threshold (< 0.497) with OR = 0.107 (95% CI: 0.015–0.748). Model selection via iterative log-likelihood ratio testing confirmed this as the optimal fit (**Table 5**). See S2 Table for characteristics of participants on both sides of the inflection point.

## Subgroup analysis

Subgroup analyses were performed to assess the possible confounding factors in the relationship between WHtR and microalbuminuria, including sex, age, BMI, hypertension, tumor presence, obesity, and diabetes (**Table 6**). Notably, significant interaction effects between WHtR and microalbuminuria emerged for hypertension (*p*-interaction = 0.0005) and BMI (*p*-interaction = 0.0007).

**Table 3. Univariate logistic regression analysis examining the risk factors for microalbuminuria.**

| Risk Factor | Statistics | Microalbuminuria (OR,95%CI, P) |
|---|---|---|
| Age(years) | 57.634±9.272 | 1.042 (1.039,1.045) < 0.00001 |
| Height(cm) | 160.406±7.967 | 0.980 (0.976,0.984) < 0.00001 |
| Weight(kg) | 63.438±10.999 | 1.001 (0.999,1.004) < 0.30985 |
| WC (cm) | 86.075±9.741 | 1.017 (1.014,1.020) < 0.00001 |
| HC (cm) | 96.947±7.755 | 1.015 (1.011,1.019) < 0.00001 |
| Scr (umol/L) | 68.540±16.961 | 1.011 (1.009,1.012) < 0.00001 |
| HDL-c (mmol/L) | 1.316±0.338 | 0.639 (0.582,0.702) < 0.00001 |
| LDL-c (mmol/L) | 2.980±0.896 | 0.901 (0.870,0.932) < 0.00001 |
| Chol(mmol/L) | 5.070±1.130 | 0.962 (0.936,0.988) < 0.00493 |
| TG (mmol/L) | 1.672±1.203 | 1.159 (1.134,1.183) < 0.00001 |
| ALT(U/L) | 17.998±13.773 | 1.005 (1.003,1.006) < 0.00001 |
| AST(U/L) | 22.433±12.452 | 1.007 (1.004,1.009) < 0.00001 |
| GGT(U/L) | 29.822±38.177 | 1.002 (1.001,1.003) < 0.00001 |
| FPG (mmol/L) | 5.986±1.717 | 1.188 (1.171,1.205) < 0.00001 |
| BPG (mmol/L) | 8.526±3.960 | 1.090 (1.083,1.097) < 0.00001 |
| HbA1c(mmol/L) | 6.094±1.049 | 1.354 (1.323,1.386) < 0.00001 |
| BMI (kg/m$^2$) | 24.598±3.516 | 1.036 (1.027,1.045) < 0.00001 |
| eGFR (ml/min*1.73m$^2$) | 93.975±19.067 | 0.986 (0.984,0.988) < 0.00001 |
| SBP (mmHg) | 131.789±20.377 | 1.020 (1.019,1.022) < 0.00001 |
| DBP (mmHg) | 77.579±10.849 | 1.021 (1.018,1.024) < 0.00001 |
| WHtR | 5.372±0.61 | 1.483 (1.410,1.559) < 0.00001 |
| Sex | | |
| Male | 11169 (33.157%) | 1.0 |
| Female | 22516 (66.843%) | 1.210 (1.132,1.293) < 0.00001 |
| Tumor | | |
| Yes | 985 (2.924%) | 1.0 |
| No | 32700 (97.076%) | 1.052 (0.876,1.264) 0.58636 |
| Diabetes Mellitus | | |
| No | 26041 (77.307%) | 1.0 |
| Yes | 7644 (22.693%) | 2.168 (2.031,2.314) < 0.00001 |
| CKD | | |
| No | 32863 (97.560%) | 1.0 |
| Yes | 822 (2.440%) | 3.457 (2.987, 4.000) < 0.00001 |
| Hypertension | | |
| No | 13412 (39.816%) | 1.0 |
| Yes | 20273 (60.184%) | 1.936 (1.809,2.071) < 0.00001 |
| Smoking | | |
| Never | 28326 (84.798%) | 1.0 |
| Sometimes | 875 (2.619%) | 0.869 (0.712,1.061) 0.16704 |
| Regular | 4203 (12.582%) | 0.860 (0.782,0.947) 0.00213 |
| Drinking | | |
| Never | 24692 (73.884%) | 1.0 |
| Sometimes | 6390 (19.120%) | 0.724 (0.666,0.787) < 0.00001 |
| Regular | 2338 (6.996%) | 0.768 (0.675,0.873.) < 0.00006 |

**Table 4. Association between waist-to-height ratio (WHtR) and microalbuminuria in multivariable-adjusted models.**

| EXPOSURE | Non-adjusted (OR, 95%CI, *p*) | Adjust I (OR, 95%CI, *p*) | Adjust II (OR, 95%CI, *p*) |
|---|---|---|---|
| WHtR | 1.483 (1.410,1.559) < 0.00001 | 1.269 (1.183,1.362) < 0.00001 | 1.173 (1.090,1.261) 0.00002 |
| WHtR quartiles | | | |
| Q1 | 1.0 | 1.0 | 1.0 |
| Q2 | 1.120 (1.020,1.229) 0.01708 | 1.018 (0.923,1.122) 0.72485 | 0.966 (0.874,1.067) 0.49699 |
| Q3 | 1.307 (1.194,1.31) < 0.00001 | 1.123 (1.105,1.243) 0.02432 | 1.028 (0.926,1.141) 0.60268 |
| Q4 | 1.829 (1.677,1.993) < 0.00001 | 1.397 (1.248,1.564) < 0.00001 | 1.242 (1.106,1.395) 0.00026 |
| WHTR quartile continuous | 1.225 (1.192,1.260) < 0.00001 | 1.121(1.081,1.162) < 0.00001 | 1.079 (1.040,1.120) 0.00006 |

Model Specifications:

Model I (Unadjusted): No adjustments.

Model II (Partially Adjusted): Adjusted for sex, blood pressure (SBP, DBP), tumor status, smoking, alcohol consumption, weight, and hip circumference (HC).

Model III (Fully Adjusted): Adjusted for all covariates from Model II along with lipid profile (HDL-c, LDL-c, TG), liver enzymes (ALT, AST), fasting plasma glucose (FPG), and estimated glomerular filtration rate (eGFR).

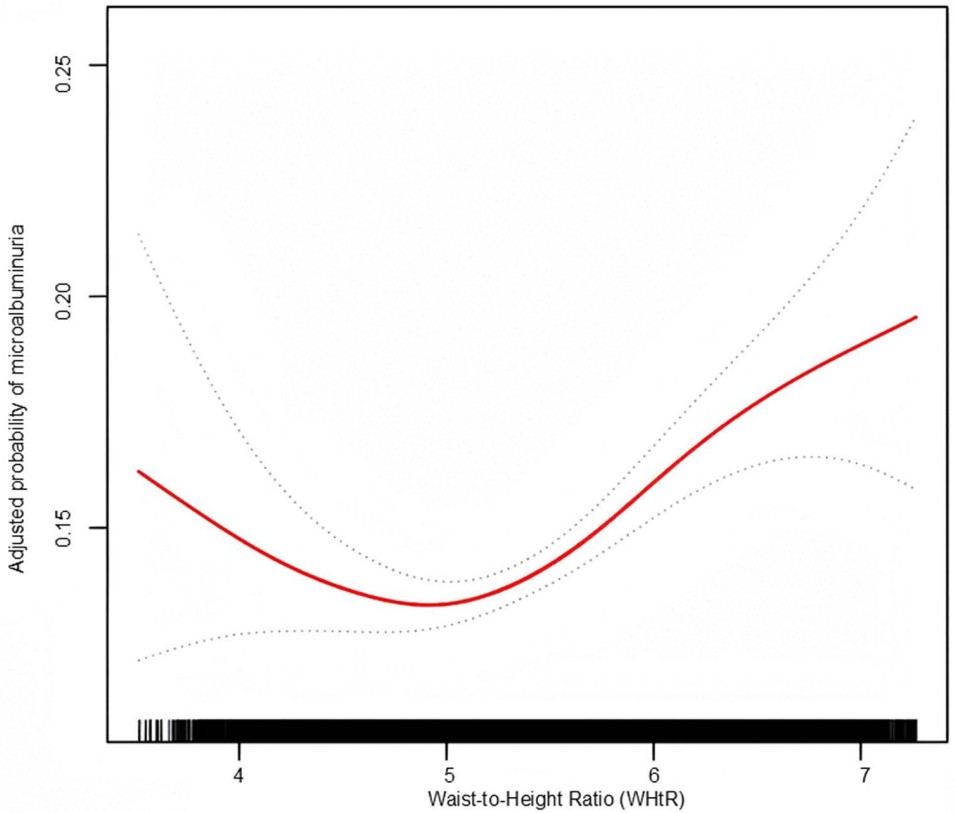

**Fig 6. Nonlinear relationship between WHtR and microalbuminuria Risk.** A U-shaped relationship was observed after full adjustment. Red solid line shows estimated association; blue dashed lines indicate 95% confidence intervals. An inflection point was identified at WHtR = 0.497.

**Table 5. Evaluation of the threshold effect of WHtR on microalbuminuria via two-piecewise cox proportional hazards regression.**

| Incident prediabetes | OR (with 95% CI) | Probability Value |
|---|---|---|
| Model I | 1.173 (1.090, 1.261) | < 0.0001 |
| Model II | | |
| WHtR infection points | 0.497 | |
| ≤0.497 | 0.107 (0.015, 0.748) | 0.0243 |
| >0.497 | 11.902 (5.141, 27.556) | < 0.0001 |
| p-Value Derived from the Log-Likelihood Ratio Test | < 0.001 | < 0.001 |

Adjustments include: sex, SBP, DBP, tumor status, smoking, alcohol intake, weight, hip circumference (HC), HDL-c, LDL-c, triglycerides (TG), liver enzymes (ALT, AST), fasting plasma glucose (FPG), and estimated glomerular filtration rate (eGFR).

OR: Odds ratio; CI: Confidence interval.

**Table 6. Analysis of subgroups for risk factors of microalbuminuria.**

| Clinical Features | Participant Count (n) | Odds Ratio (OR) [95% Confidence Interval (CI)] | Probability Value (P) | Interaction p-Value (p-interaction) |
|---|---|---|---|---|
| Age | | | | 0.4415 |
| ≤ 60(years) | 21981 | 0.995 (0.346,2.861) | 0.9927 | |
| > 60(years) | 11274 | 1.7520 (0.626,4.904) | 0.2859 | |
| Sex | | | | 0.2557 |
| Male | 11052 | 1.269 (1.100,1.462) | 0.0001 | |
| Female | 22203 | 1.159 (1.070,1.256) | 0.0003 | |
| Diabetes | | | | 0.6845 |
| No | 25707 | 1.143 (1.047,1.249) | 0.0029 | |
| Yes | 7548 | 1.181 (1.038,1.342) | 0.0112 | |
| Hyper-tension | | | | 0.0005 |
| Yes | 13206 | 1.045 (0.915,1.193) | 0.5182 | |
| No | 20049 | 1.374 (1.263,1.494) | < 0.0001 | |
| Obesity | | | | 0.0105 |
| Yes | 4940 | 1.409 (1.187,1.673) | < 0.0001 | |
| No | 28315 | 1.116 (1.030,1.208) | 0.0071 | |
| Tumor | | | | 0.5448 |
| Yes | 972 | 1.344 (0.859,2.104) | 0.1954 | |
| No | 32283 | 1.169 (1.086,1.258) | <0.0001 | |
| BMI | | | | 0.0007 |
| < 18.5 | 772 | 0.878 (0.507,1.520) | 0.6424 | |
| ≥ 18.5<24 | 14365 | 0.967 (0.852,1.098) | 0.6059 | |
| ≥ 24,<28 | 13178 | 1.322 (1.158,1.510) | < 0.0001 | |
| ≥ 28 | 4940 | 1.426 (1.173,1.734) | 0.0004 | |

Note 1: The model was adjusted for age, sex, body mass index (BMI), and the history of diabetes, hypertension, obesity, and tumors.

Note 2: In every case, the model does not account for adjustments based on the stratification variable.

OR: Odds ratio; CI: Confidence interval.

When stratified by hypertension status, WHtR showed a stronger association with microalbuminuria in non-hypertensive individuals (OR = 1.374, 95% CI: 1.263–1.494; $p < 0.0001$), while the association was attenuated and non-significant among participants with hypertension (OR = 1.045, $p = 0.5182$). Similarly, WHtR exhibited a BMI-dependent risk pattern. The strongest association with microalbuminuria in individuals with BMI ≥ 28 kg/m$^2$ (OR = 1.426, 95% CI: 1.173–1.734), whereas the association was not significant among normal-weight participants (BMI 18.5–24 kg/m$^2$; OR = 0.967, $p = 0.6059$). These findings suggested that both elevated BMI and the absence of hypertension potentiate the deleterious renal effects of central adiposity, pointing to a potential threshold effect in the higher BMI stratum.

No significant interactions were identified for sex, diabetes, or tumor history (all $p$-interaction > 0.05). Although the age-stratified OR was higher in participants > 60 years (1.752 vs 0.995), the interaction effect remained non-significant ($p$-interaction = 0.4415), warranting cautious interpretation.

## Discussion

This cross-sectional study examined the relationship between WHtR and microalbuminuria in a representative Chinese population and revealed a statistically significant, nonlinear U-shaped association. Using generalized additive models, we identified a threshold WHtR value of 0.497, with contrasting associations on either side of this point: values below the threshold were associated with decreased risk of microalbuminuria, while values above the threshold were strongly linked to increased risk. Subgroup analyses further clarified that this association was most pronounced among individuals with obesity (BMI ≥ 28 kg/m$^2$) and those without hypertension. These findings suggest that obesity may amplify the metabolic and inflammatory burden on renal function, potentially enhancing susceptibility to microvascular damage. The attenuated association in hypertensive individuals may reflect overlapping renal injury mechanisms that obscure WHtR's independent contribution, or may be influenced by antihypertensive medication use or duration of disease.

Previous studies have demonstrated associations between central obesity and renal dysfunction. Obesity has been shown to correlate with elevated albumin excretion rate [20], and multiple studies have identified obesity as a significant risk factor for decreased estimated glomerular filtration rate and chronic kidney disease [21–23,35], supporting a biological linkage between abdominal adiposity and the pathophysiological mechanisms underlying microalbuminuria [24–28]. Furthermore, indices of visceral adiposity, such as waist circumference and waist-to-hip ratio, have been demonstrated to outperform BMI as predictors of end-stage renal disease [29]. Consistent with these findings, the REVEND study revealed that individuals exhibiting central obesity faced an elevated risk of microalbuminuria relative to those with a predominantly peripheral fat distribution pattern [30].

The relationship between adiposity indices and albumin-to-creatinine ratio appears to vary across ethnic populations [31]. Studies conducted in Surinamese and Dutch populations [32], Japanese adults [33], and Chinese individuals [34] have all reported associations between various obesity measures and elevated ACR, though the specific indices showing significance differed across populations. These variations highlight the importance of population-specific investigations and suggest that WHtR may be a particularly relevant metric in Chinese adults.

Interestingly, a German investigation described a U-shaped correlation between obesity and ACR, indicating that the likelihood of ACR elevation was increased in both underweight and obese populations [35]. Our results corroborate and extend these findings, demonstrating a statistically significant U-shaped correlation between WHtR and microalbuminuria in a large Chinese cohort. The observed association at the lower WHtR range may reflect the unique characteristics of our study population, including a higher prevalence of sarcopenic obesity compared to reference studies [36]. Methodological variations in microalbuminuria assessment and the absence of standardized cut-offs for low WHtR in current guidelines [37,38] likely also contributed to this finding. Our investigation provides a novel perspective on the relationship between WHtR and microalbuminuria, positioning WHtR as a clinically accessible, modifiable indicator for early intervention to prevent microalbuminuria, particularly in individuals with obesity and without overt hypertension. The nonlinear relationship remained robust after adjusting for a comprehensive panel of covariates, underscoring its clinical relevance.

The biological plausibility of this association is supported by the role of adipose tissue as an active endocrine organ. Adipose tissue secretes numerous bioactive molecules that modulate blood pressure regulation, lipid metabolism, and glucose homeostasis [39,40]. The association between WHtR and microalbuminuria may be mediated by adipose-driven inflammation and insulin resistance, both of which contribute to endothelial dysfunction and glomerular hyperfiltration [41–43]. Adipokines such as tumor necrosis factor-α and interleukin-6 are plausible mechanistic intermediates linking abdominal adiposity to kidney damage. At the lower end of WHtR, the increased risk may be related to sarcopenia or chronic undernutrition, which can also lead to systemic inflammation and endothelial dysfunction.

Several key strengths characterize our study. First, the large sample size derived from the general population across eight regions in China enhances the representativeness and statistical power of our results. Second, we identified a non-linear U-shaped association between WHtR and microalbuminuria, an aspect seldom addressed in prior research. Third, we minimized potential confounding through rigorous multivariable adjustments. Finally, comprehensive subgroup analyses confirmed the robustness and consistency of our findings across diverse population strata.

Nonetheless, our study has certain limitations. Most importantly, the cross-sectional design inherently precludes any causal inferences regarding the relationship between WHtR and microalbuminuria. While we identified a robust U-shaped association, we cannot establish the temporal sequence—that is, whether low or high WHtR precedes the development of microalbuminuria, or whether reverse causality (e.g., underlying kidney disease leading to changes in body composition) or unmeasured confounding explains this relationship. Second, reliance on a single spot urine sample for ACR measurement may introduce intra-individual variability and potential misclassification, although this method is widely accepted in large-scale epidemiological studies. Third, while we adjusted for a comprehensive panel of confounders, residual confounding from unmeasured variables—such as detailed dietary habits, physical activity levels, socioeconomic status, and genetic predisposition—cannot be fully excluded. Fourth, the generalizability of our findings to other ethnic populations requires further validation, as the study was conducted exclusively in Chinese adults. Therefore, future prospective longitudinal cohort studies with repeated ACR measurements, longer follow-up periods, and broader inclusion of potential confounders are urgently warranted to validate our findings and establish causality.

## Conclusion

This study investigated the relationship between WHtR and microalbuminuria in a large, diverse Chinese population. Our analyses revealed a statistically significant non-linear, U-shaped dose-response association, with both low and high WHtR values linked to elevated risk of microalbuminuria. This relationship persisted after comprehensive multivariable adjustment, underscoring its robustmess. Notably, subgroup analyses indicated that BMI and hypertension status may modify this association, suggesting potential thresholds where WHtR may exert stronger effects.

These findings also emphasized abdominal obesity as a clinically relevant determinant of renal health and supported the use of WHtR as a simple, noninvasive marker for identifying individuals at risk of renal dysfunction. Tailored prevention strategies may be beneficial to address differing risk profiles associated with different WHtR phenotypes. However, given the cross-sectional nature of this study, these findings should be interpreted with caution regarding causality. Future prospective longitudinal studies are needed to confirm the observed U-shaped association, establish the temporal relationship between WHtR and microalbuminuria, and evaluate whether interventions targeting WHtR can effectively reduce the risk of renal dysfunction.

## Supporting information

**S1 Table. Collinearity diagnostics steps.** This table presents the variance inflation factor (VIF) values for all covariates included in the multivariable models.
(DOCX)

**S2 Table. Characteristics of participants on both sides of the inflection point.** This table compares demographic, anthropometric, biochemical, and clinical variables between participants with WHtR < 0.497 (n = 8,643) and those with WHtR ≥ 0.497 (n = 25,042).
(DOCX)

**S1 File. Raw dataset.** This Excel file contains the anonymized participant-level data used for all statistical analyses.
(XLSX)

## Acknowledgments

The present secondary analysis leverages data and analytical frameworks derived from a prior investigation conducted by Ye et al. (2019), which systematically examined the relationships between self-reported sleep duration, daytime napping behavior, renal hyperfiltration, and microalbuminuria among a cohort of healthy Chinese individuals. The foundational study was published in the peer-reviewed journal *PLOS ONE* (accessible via https://doi.org/10.1371/journal.pone.0214776). We would like to acknowledge the contributions of all authors involved in the original research for their essential work.

## Author contributions

**Data curation:** Xia Huang, Xiaofang He.

**Formal analysis:** Xia Huang, Haofei Hu, Zhichao Zhang, Xue Zhang, Hao Yuan.

**Funding acquisition:** Xiuqin Li.

**Supervision:** Lirong Tu, Feiyuan Liu, Heping Zhang, Yongcheng He.

**Writing – original draft:** Xia Huang.

**Writing – review & editing:** Xia Huang, Lishu He.

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
