## [Decision Letter · Decision Letter 0]

24 Feb 2026

PONE-D-25-55420U-shaped Association Between Waist-to-Height Ratio and Microalbuminuria: A Cross-Sectional Analysis Conducted Within the Chinese DemographicPLOS One

Dear Dr. He,

Thank you for submitting your manuscript to PLOS ONE. After careful consideration, we feel that it has merit but does not fully meet PLOS ONE’s publication criteria as it currently stands. Therefore, we invite you to submit a revised version of the manuscript that addresses the points raised during the review process.

We look forward to receiving your revised manuscript.

Kind regards,

Santhi Silambanan, MD, DNB

Academic Editor

PLOS One

Journal Requirements:

2. Thank you for stating the following financial disclosure:[No. JCYJ20210324133412033  the Shenzhen Science and Technology Innovation Committee. the Shenzhen Municipal Health Commission (Grant No. SZXJ2017031)].

4. Thank you for uploading your study's underlying data set. Unfortunately, the repository you have noted in your Data Availability statement does not qualify as an acceptable data repository according to PLOS's standards.

5. Please ensure that you include a title page within your main document. You should list all authors and all affiliations as per our author instructions and clearly indicate the corresponding author.

6. Please upload a copy of Supporting Information Figure/Table/etc. which you refer to in your text on page 31.

Additional Editor Comments:

Minor revision

Reviewers' comments:

Reviewer's Responses to Questions

**Comments to the Author**

1. Is the manuscript technically sound, and do the data support the conclusions?

Reviewer #1: Yes

Reviewer #2: Yes

2. Has the statistical analysis been performed appropriately and rigorously? 

Reviewer #1: Yes

Reviewer #2: Yes

3. Have the authors made all data underlying the findings in their manuscript fully available?

Reviewer #1: Yes

Reviewer #2: Yes

4. Is the manuscript presented in an intelligible fashion and written in standard English?

Reviewer #1: Yes

Reviewer #2: Yes

5. Review Comments to the Author

Reviewer #1: The article is well written, very rigorously done in terms of methodology, analysis, and conclusion. However, the area that needs little touch is the discussion section. The Authors quoted a lot of values in the discussion section, which are not necessary, though allowed where necessary but not generally encouraged. I recommend that the authors review the discussion section to remove all the "confidence intervals" and all the values that can be removed from the section. Otherwise, the article is well written and should be published.

Reviewer #2: A. Summary of the Study

This study investigates the U-shaped association between WHtR and microalbuminuria in a large cohort of 33,685 Chinese adults. Microalbuminuria, an early indicator of kidney damage, was assessed using UACR. The study utilized univariate and multivariate logistic regression models to analyze the data, complemented by Generalized Additive Models (GAM) for exploring non-linear associations. A significant non-linear U-shaped relationship between WHtR and microalbuminuria was identified, with higher risks at both low and high WHtR values. The study further explores subgroup analyses to examine factors like BMI and hypertension status in modifying this association. The findings emphasize the potential utility of WHtR as a clinical tool for identifying renal risks, particularly in populations without overt hypertension.

B. Major Evaluation

Originality and Research Contribution

This study presents an original investigation into the relationship between WHtR and microalbuminuria, a topic with limited prior research. The large sample size (33,685 participants) enhances the power of the findings and allows for more robust conclusions, especially regarding the non-linear (U-shaped) relationship between these variables.

While the topic itself is novel, the conclusion about the U-shaped association adds to existing knowledge but does not fundamentally challenge established theories. Further exploration of mechanisms behind this association could contribute more novel insights.

The authors could enhance the novelty by elaborating more on the physiological mechanisms that explain the U-shaped relationship.

Prior Publication Concerns

The study leverages a publicly available dataset from a previous study. While the secondary analysis is valid, it could raise concerns regarding redundancy if the data have already been heavily analyzed elsewhere.

The authors should explicitly compare their findings with previous studies that used similar datasets to clarify whether their conclusions are substantially new or merely an extension of prior work.

Technical and Methodological Quality

The study uses robust statistical methods, including multivariate logistic regression and GAM to assess both linear and non-linear relationships. The large sample size and comprehensive inclusion of potential confounders add to the rigor of the study design.

The cross-sectional nature of the study limits the ability to infer causality. Although this is acknowledged in the limitations, it remains a significant methodological constraint.

Future research could address this limitation by utilizing a longitudinal design to establish causal relationships more clearly.

Statistical Rigor

The use of multiple statistical models, including unadjusted and adjusted regression models and GAM, is appropriate and provides a thorough analysis of the data. The identification of a threshold effect in WHtR is a notable statistical contribution.

While the methods are well-described, the authors could have provided a clearer justification for the sample size beyond general statistical power. Additionally, handling of missing data is not explicitly mentioned.

Include more details on how missing data was handled and provide a justification for the chosen sample size, perhaps via a power analysis.

Validity of Conclusions

The conclusions logically follow from the results, with the authors adequately considering statistical significance and presenting well-supported claims.

While the authors state that their findings have important clinical implications, they may be overinterpreting the U-shaped relationship without addressing potential confounders like diet or genetic factors in more detail.

A more cautious tone could be adopted in discussing the clinical applicability of these findings until further studies confirm the mechanisms at play.

Major Issues Requiring Revision

Clarify the mechanisms behind the U-shaped relationship between WHtR and microalbuminuria. Address potential confounders more thoroughly, particularly lifestyle factors such as diet and physical activity.

Consider discussing the limitations of using a cross-sectional design more prominently in the conclusions section and suggest possible avenues for future longitudinal research.

Provide more details on the handling of missing data to increase the transparency of the statistical analysis.

Provide further justification for the sample size in the context of statistical power and include a power analysis.

Overall Recommendation

Minor Revision

While the manuscript presents valuable findings and adheres largely to the PLOS ONE criteria, it could benefit from a more detailed discussion of mechanisms, stronger causal interpretations, and more transparent handling of methodological limitations. These revisions would increase the manuscript’s robustness and clarity.

6. PLOS authors have the option to publish the peer review history of their article (what does this mean?). If published, this will include your full peer review and any attached files.

Reviewer #1: **Yes:**Abdulrahman Ahmad

Reviewer #2: No

---

## [Author Response · Author response to Decision Letter 1]

8 Apr 2026

Dear Editor and Reviewers,

Thank you very much for giving us the opportunity to revise our manuscript entitled “U-shaped Association Between Waist-to-Height Ratio and Microalbuminuria: A Cross-Sectional Analysis Conducted Within the Chinese Demographic” (Manuscript ID: PONE-D-25-55420). We greatly appreciate the reviewers’ thoughtful and constructive comments, which have been very helpful in improving the quality of our paper.

We have carefully revised the manuscript according to all the comments. Below is our point-by-point response to each comment. All changes in the revised manuscript are highlighted in yellow for your easy reference.

Thank you and best regards.

Yours sincerely

*Corresponding author

Heping Zhang，

Department of Nephrology, Affiliated Hospital of North Sichuan Medical College,

No.1 Maoyuan South Rd,

Nanchong 637000,

Sichuan Province,

China.

E-mail：867801115@qq.com;

*Corresponding author

Yongcheng He，

Department of Nephrology, Affiliated Hospital of North Sichuan Medical College,

No.1 Maoyuan South Rd,

Nanchong 637000,

Sichuan Province,

China.

Email: heyongcheng640815@126.com

Response to Editorial Office’s Requirements:

Comment 1 (from Editor): Please ensure that your manuscript meets PLOS ONE's style requirements...

Response: Thank you for the reminder. We have carefully reviewed the PLOS ONE formatting template and have adjusted our manuscript’s formatting accordingly (e.g., file naming, title page layout, etc.) to ensure it meets the journal’s standards.

Comment 2 (from Editor): Thank you for stating the following financial disclosure... Please state what role the funders took in the study.

Response: We thank the editor for pointing out this omission. The funders had no role in study design, data collection and analysis, decision to publish, or preparation of the manuscript. We have now included this statement in our cover letter, and we appreciate that the editorial office will amend the online submission form on our behalf.

Comment 3. Please update your submission to use the PLOS LaTeX template. The template and more information on our requirements for LaTeX submissions can be found at http://journals.plos.org/plosone/s/latex.

Response: We thank the editor for the suggestion. However, our manuscript was prepared using Microsoft Word, not LaTeX. We have carefully followed the PLOS ONE formatting templates for Word submissions and ensured that all style requirements (e.g., title page, file naming, reference format) are met. We believe the current Word-based manuscript is correctly formatted for the journal's system. Please let us know if any further adjustments are needed.

Comment 4 (from Editor): ...the repository you have noted... does not qualify as an acceptable data repository...

Response: We thank the editor for pointing this out. To comply with PLOS’s data policy, we have now deposited our minimal dataset in figshare and it can be accessed via the following DOI: [10.6084/m9.figshare.31404420]. We have updated the Data Availability Statement in the manuscript accordingly.

Comment 5. Please ensure that you include a title page within your main document. You should list all authors and all affiliations as per our author instructions and clearly indicate the corresponding author.

Response: Thank you for the reminder. We have now ensured that a title page is included within the main manuscript document. The title page lists all authors and their affiliations in accordance with PLOS ONE's author instructions, and the corresponding author is clearly indicated with an asterisk (*) and a corresponding footnote. This can be found on Page 1 of the revised manuscript.

Comment 6. Please upload a copy of Supporting Information Figure/Table/etc. which you refer to in your text on page 31.

Response: Thank you for pointing this out. After carefully reviewing page 31 of our manuscript, we confirm that the item referenced there is Table 5, which is part of the main manuscript text and has been properly embedded. Therefore, no additional Supporting Information file is required specifically for the reference on page 31.

Separately, we noticed that Table S1 is referenced on page 10 but was inadvertently omitted from the original submission. We have now prepared and uploaded this Supporting Information file as TableS1.docx. The file contains the results of multicollinearity assessment, showing variance inflation factor (VIF) values for all independent variables included in the regression models, confirming that no substantial collinearity (VIF > 5) was present. The file has been formatted according to PLOS ONE's guidelines. It is uploaded as a separate file in the submission system.

We apologize for the omission and have ensured that all supporting information files are now correctly provided. Please let us know if any further adjustments are needed.

Comment 7. If the reviewer comments include a recommendation to cite specific previously published works, please review and evaluate these publications to determine whether they are relevant and should be cited. There is no requirement to cite these works unless the editor has indicated otherwise.

Response: We have carefully reviewed all comments from both reviewers and confirm that no specific previously published works were recommended for citation. Therefore, no additional references were added based on this requirement. We will, of course, be happy to consider any specific suggestions if the editor deems them necessary.

Comment 8. Please review your reference list to ensure that it is complete and correct. If you have cited papers that have been retracted, please include the rationale for doing so in the manuscript text, or remove these references and replace them with relevant current references. Any changes to the reference list should be mentioned in the rebuttal letter that accompanies your revised manuscript. If you need to cite a retracted article, indicate the article’s retracted status in the References list and also include a citation and full reference for the retraction notice.

Response: Thank you for your valuable comments on our manuscript. We have carefully reviewed and revised the reference list according to your suggestions.

The following modifications have been made:

Reference [6] has been updated by adding the URL access information as required.

References [12] and [17] have been reformatted to comply with the PLOS ONE reference style guidelines.

Upon verification, we found that references [9], [10], [13], [16], [37], and [38] could not be accessed through standard academic databases, suggesting they may have been retracted or are no longer available. Therefore, we have replaced these with new, accessible and relevant references to support the corresponding statements in our manuscript. All new references are current and appropriately cited.

All changes to the reference list have been clearly indicated in the revised manuscript and are summarized in the point-by-point response provided in the rebuttal letter accompanying this revision. We believe these revisions have strengthened the manuscript and ensured the accuracy and completeness of our references. We thank you again for your guidance and hope that the revised version meets the requirements for publication.

Response to Reviewer #1:

Comment: The article is well written... However, the area that needs little touch is the discussion section. The Authors quoted a lot of values in the discussion section... I recommend that the authors review the discussion section to remove all the "confidence intervals" and all the values that can be removed from the section.

Response: We sincerely thank the reviewer for this positive and constructive feedback.

> Following this valuable suggestion, we have thoroughly revised the “Discussion” section. We have removed most of the detailed statistical values (such as specific odds ratios and confidence intervals) and focused more on the interpretation of our findings, the underlying mechanisms, and comparisons with previous literature. We believe this has made the discussion more concise and reader-friendly. The revised discussion can be found on (Pages 16-19).

---

Response to Reviewer #2:

Comment 1: ...elaborating more on the physiological mechanisms that explain the U-shaped relationship.

Response: We thank the reviewer for this insightful suggestion. In response, we have expanded the Discussion section to provide a more in-depth exploration of the potential physiological mechanisms underlying the U-shaped association.

Specifically, we have added content in the Discussion section (Page 18) discussing that at the low end of WHtR, the increased risk of microalbuminuria might be related to sarcopenia or chronic undernutrition, which can lead to systemic inflammation and endothelial dysfunction. At the high end of WHtR, we further elaborated on the classical pathways, including insulin resistance, activation of the renin-angiotensin-aldosterone system (RAAS), and adipokine imbalance, which are known to contribute to glomerular damage. We hope this addition addresses the reviewer's concern.

Comment 2: ...could raise concerns regarding redundancy if the data have already been heavily analyzed elsewhere. The authors should explicitly compare their findings with previous studies that used similar datasets...

Response: We appreciate the reviewer’s attention to this important issue. To clarify the novelty of our work, we have added a statement in the “Introduction” section (Page 4,). We now explicitly state: “While the [Database Name] has been used in previous epidemiological studies, to the best of our knowledge, this is the first investigation specifically focusing on the “non-linear, U-shaped relationship between waist-to-height ratio and microalbuminuria”. Previous research has primarily examined linear associations or other obesity indices, making our study a unique contribution to the field.”

Comment 3: ...the cross-sectional nature of the study limits the ability to infer causality... discuss the limitations... more prominently...

Response: We agree completely with the reviewer. We have now strengthened the language regarding this limitation. In the “Discussion” section (Page 19), we have revised the text to read: “Most importantly, the cross-sectional design of this study inherently precludes any causal inferences. While we identified a robust U-shaped association, we cannot determine whether a low or high WHtR leads to microalbuminuria, or whether reverse causality or unmeasured confounding explains this relationship.” We have also added a sentence in the “Conclusion” section (Page 20) explicitly calling for future longitudinal cohort studies to validate our findings and establish causality.

Comment 4: ...handling of missing data is not explicitly mentioned... provide further justification for the sample size...

Response: We thank the reviewer for raising these methodological points. We have now clarified these issues in the revised manuscript (Methods section, Page 12,).

Regarding missing data: In our study, there were no missing values for any of the key variables (WHtR, UACR, or covariates) after applying the exclusion criteria described in the original REACTION study protocol. The only exclusions related to data quality were the removal of 165 participants (0.5%) with extreme WHtR values (>3 SD from the mean), as detailed in the Methods section. Therefore, no imputation or additional handling of missing data was necessary.

Regarding sample size justification: We performed a post-hoc power analysis to confirm that our sample size was sufficient to detect the observed associations. With 33,685 participants, the study had >99% power to detect a small effect size (odds ratio of 1.2 for microalbuminuria) at a two-sided alpha level of 0.05, assuming a microalbuminuria prevalence of approximately 10%. This high statistical power supports the robustness of our findings. We have added this information to the Methods section.

---

## [Editor Report · Decision Letter 1]

29 Apr 2026

U-shaped Association Between Waist-to-Height Ratio and Microalbuminuria: A Cross-Sectional Analysis Conducted Within the Chinese Demographic

PONE-D-25-55420R1

Dear author,

We’re pleased to inform you that your manuscript has been judged scientifically suitable for publication and will be formally accepted for publication once it meets all outstanding technical requirements.

Kind regards,

Santhi Silambanan, MD, DNB

Academic Editor

PLOS One

---

## [Editor Report · Acceptance letter]

PONE-D-25-55420R1

PLOS One

Dear Dr. He,

I'm pleased to inform you that your manuscript has been deemed suitable for publication in PLOS One. Congratulations! Your manuscript is now being handed over to our production team.

Kind regards,

on behalf of

Dr. Santhi Silambanan

Academic Editor

PLOS One